# The Use of Data from the Parkinson’s KinetiGraph to Identify Potential Candidates for Device Assisted Therapies

**DOI:** 10.3390/s19102241

**Published:** 2019-05-15

**Authors:** Hamid Khodakarami, Parisa Farzanehfar, Malcolm Horne

**Affiliations:** 1Global Kinetics Corporation, 31 Queens St., Melbourne 3000, Australia; Hamid.Khodakarami@globalkineticscorp.com; 2Florey Institute of Neuroscience & Mental Health, The University of Melbourne, Parkville 3010, Australia; parisa.farzanehfar@nh.org.au; 3St Vincent’s Hospital, Fitzroy 3065, Australia

**Keywords:** deep brain stimulation, device assisted therapies, objective measurement, wearing off, bradykinesia, dyskinesia

## Abstract

Device-assisted therapies (DAT) benefit people with Parkinsons Disease (PwP) but many referrals for DAT are unsuitable or too late, and a screening tool to aid in identifying candidates would be helpful. This study aimed to produce such a screening tool by building a classifier that models specialist identification of suitable DAT candidates. To our knowledge, this is the first objective decision tool for managing DAT referral. Subjects were randomly assigned to either a construction set (n = 112, to train, develop, cross validate, and then evaluate the classifier’s performance) or to a test set (n = 60 to test the fully specified classifier), resulting in a sensitivity and specificity of 89% and 86.6%, respectively. The classifier’s performance was then assessed in PwP who underwent deep brain stimulation (n = 31), were managed in a non-specialist clinic (n = 81) or in PwP in the first five years from diagnosis (n = 22). The classifier identified 87%, 92%, and 100% of the candidates referred for DAT in each of the above clinical settings, respectively. Furthermore, the classifier score changed appropriately when therapeutic intervention resolved troublesome fluctuations or dyskinesia that would otherwise have required DAT. This study suggests that information from objective measurement could improve timely referral for DAT.

## 1. Introduction

Parkinson’s Disease (PD) is the second most common neurodegenerative disease, affecting over 6 million people. It affects movement, cognition, the autonomic nervous system, and causes neuropsychiatry. Clinical interest focuses on disturbances of movement because they cause significant disability, they can be treated, and they dominate the early years after diagnosis. The key diagnostic feature of PD is the slowness of movement, known as bradykinesia, and this is caused by a loss of dopamine transmission in the striatum. Bradykinesia can be treated effectively by dopaminergic medications including levodopa. The first few years of Parkinson’s Disease (PD) respond well [1,2] but the duration of symptomatic benefit derived from each levodopa dose begins to shorten in ~50% of people with PD (PwP) after two years of disease and ~70% of PwP eventually experience this loss of therapeutic benefit, known as “wearing-off” or “off” periods or fluctuations [3,4]. At about the same time, involuntary movements at the peak of the levodopa dose, known as dyskinesia, begin to emerge. Much of the therapeutic effort is directed at reducing these fluctuations between bradykinesia and dyskinesia as they lead to disability and impaired quality of life. Initially, this can be achieved by adjusting oral therapies but device-assisted therapies (DAT) such as deep brain stimulation provides superior results in many PwP. 

Despite broad consensus as to the criteria for selecting DAT candidates [5,6], non-specialists have difficulty in recognizing these criteria. Many PwP in whom fluctuations are emerging are managed by non-specialists and consequently, suitable DAT candidates are not referred in a timely manner [7]. For example, the timing for deep brain stimulation (DBS) is important because there is a window of optimum benefit [8,9], and delay means that suitable candidates may have shorter benefit from DBS or worse still, miss out entirely. As many as 67% of patients referred for DBS are unsuitable for the procedure [5,10] yet only 1% of people with PD receive DBS [11], even though as many as 20% may, in fact, be eligible [6]. One reason is that fluctuations, a key indication for DBS and other DAT [12], are frequently overlooked by both patient and clinician [13,14,15]. The motor indication for consideration of any DAT is similar [16], consisting of troublesome periods of bradykinesia (“off” periods) or dyskinesia that cannot be addressed by optimal deployment of oral therapies. Responsiveness to levodopa is important, but often an increased number of daily levodopa doses are required. Unquestionably, age and cognition influence the type and threshold for DAT, but these should be addressed at the specialist referral center rather than a reason to delay referral when oral therapies do not address troublesome “off-times” and or dyskinesia. Thus, a screening aid could have a role in ensuring that suitable candidates are referred to specialist centers for full consideration of all the factors that influence suitability for DAT without burdening these centers with too many unsuitable cases [5,10]. Our interest in this study was whether a classifier that provided this screening aid could be built using recently developed wearable sensors for PD [17,18]. The assessment of PwP, including suitability for DAT, is currently based entirely on clinical skills. The recent developments in objective measurements of the motor features of PD [17,18] raise the possibility of using data from wearable sensors to build an instrumented classifier to assist in the detection of DAT candidates. To our knowledge, our previous pilot study has been the only attempt to do this [19]. 

The Parkinson’s KinetiGraph (PKG), described further in the method section, is a wearable sensor system that provides objective scores of the motor features of PD, including bradykinesia, dyskinesia, and fluctuations. Preliminary data suggest that it substantially improves the recognition of fluctuations [15]. This should lend itself to a machine learning approach to recognize DAT candidates. In a previous pilot study [19], 36 people with PD (PwP) were classified on motor grounds as either being DBS candidates or as unsuitable candidates by clinicians who were expert in DBS. The information from the PKG obtained at the time of classification was used to model this decision and build a predictive score [19]. Although this score had high sensitivity and specificity on its original training set, it was not formally tested in a re-test cohort. Furthermore, only small differences in the score separated people who were DBS candidates on motor grounds from those that were unsuitable. A score that addressed these issues by providing a likelihood or risk (in a statistical sense) of requiring DAT might better address the needs of the non-specialist referrer. 

In the study reported here, a new score (a DAT classifier score) that predicted the likelihood of a PwP being a DAT candidate was developed. Establishing whether a PwP should be considered for DAT (or not) is a classification problem where the benchmark is the opinion of the expert clinician. There are well established processes for building classifiers that automate human decision making. We outline the steps here to aid the reader who is less familiar with these processes and to foreshadow the results in this paper.

The first step in building a classifier is to identify construction and test sets (of PwP in our case). The construction set was used to train, develop, cross-validate, and evaluate the performance of the fully specified classifier, which was then re-tested on the test set. Ideally, the construction and test sets should be randomly selected from the same integral population of PwP. The second key step is the choice and refinement of PKG variables. While there are many options, we chose intuitively selected candidate PKG variables, based on information from the literature (see reference [16] for a review) and from experience gained in developing the earlier classifier described above [19]. We then used statistical methods (joint mutual information) to determine which of these parameters were most important in carrying information reflected in the classification of a subject as meeting the criteria for DAT (criteria positive (CP)) or not meeting the criteria for DAT (criteria negative (CN)). This shortened the list of candidate PKG variables to those containing the most non-redundant relevant information and descriptive information about the CP and CN classes. The next step was to build a model that used these parameters to predict the clinical classification. The accuracy of the model was assessed using the technique of cross validation, which uses sub-samples of the construction set to assess the accuracy of the prediction using the area under a receiver operating characteristic (ROC) curve. For the design of the classification model pipeline, k-fold cross validation was performed using the construction set, and the area under the curve of the receiver operating characteristic (ROC) was used as the performance criteria. ROC is an appropriate measure here because the classes are balanced. Different elements throughout the pipeline were iteratively modified until the best performance on the ROC curve was obtained (judged by AUC and sensitivity vs. specificity). Only after the performance of the model was optimal on the construction set (described in detail in the Results section), it was tested on the separate test set with the expectation of achieving low variance suggesting generalizability of the model to any unseen data. 

When this stage was reached, we could say at one level, that the classifier had been validated. However, further steps were required to obtain insights into the clinical validity and limitations of the classifier. Any classifier will have errors and before using it in clinical decision making, the nature of these errors should be understood. Thus, there is value in examining cases that were either incorrectly overlooked as DAT candidates or incorrectly identified as DAT candidates. Understanding the reasons for false negatives and positives not only informs clinicians in using the classifier but also aids the development of the classifier in the future. Conceptually, a PwP with excessive periods of bradykinesia or dyskinesia would be classified (by the classifier) as “suitable” for DAT, yet doesn’t address the question of whether the excessive periods of bradykinesia or dyskinesia can be resolved by manipulation of oral therapies or whether DAT is required. This is encapsulated by the general commentary that DAT is recommended when there are excessive periods of bradykinesia that cannot be addressed by manipulating oral therapies. The findings suggest that in the likely real-world practice, a clinician using a support algorithm would recommend DAT once they were confident that the score could not be improved by manipulating oral therapy. We tested this by examining the management of a population of PWP to see how many PwP with excessive periods of bradykinesia or dyskinesia that would otherwise indicate suitability for DAT, could be improved by manipulating oral therapies. The DAT classifier score produced using a machine learning program shows promise as a tool to guide clinicians as to when PwP have reached a point when referral to an expert center is timely. 

## 2. Materials and Methods

The data used in this study were collected from 100 subjects attending Movement Disorder Clinics in Australia for consideration of DAT or to optimize their PD treatment and from 72 subjects recruited into various previous studies. This study was carried out in accordance with the guidelines issued by the National Health and Medical Research Council of Australia for Ethical Conduct in Human Research (2007, and updated May 2015). Approval to use this data was provided by St Vincent’s Hospital Melbourne Human Research & Ethics Committee (Approval Number HREC/12/SVHM/11). All participants had provided written informed consent prior to participation in accordance with the Declaration of Helsinki. 

All patients were assessed by Movement Disorder specialists for their suitability for DAT and classed according to whether or not there were troublesome “off” periods or dyskinesia that could not be addressed by deployment of oral therapies [16]. Clinicians were asked explicitly not to consider non-motor factors in this decision. They were thus grouped as clinical “criteria positive (CP)” or clinical “criteria negative” (CN).
Met clinical criteria for DAT (criteria positive (CP)). Typically these were PwP requiring frequent doses of levodopa with troublesome ‘off’ periods (>1–2 h/day) and or dyskinesia [16].Did not meet the clinical criteria for DAT (criteria negative (CN)). Typically these were PwP without a history of severe motor fluctuations or dyskinesia or if these motor fluctuations or dyskinesia had been resolved by optimization of oral therapies.

PwP in the study were also assessed using the Movement Disorder Society-Sponsored Unified Parkinson’s Disease Rating Scale (MDS-UPDRS) [20].

### 2.1. The PKG System

The PKG system consists of a wrist-worn data logger (the PKG logger), a series of algorithms that produce data points every two minutes and a series of graphs and scores that synthesize this data into a clinically useful format known as the PKG. The PKG system [21] was the first system to receive clearance from the FDA to measure bradykinesia, and also has indications for measuring dyskinesia, tremor, and sleep. It is the only commercially available system providing a continuous, objective, and ambulatory assessment of bradykinesia.

The logger is a smart watch that is worn on the most affected wrist. It weighs 35 g and contains a rechargeable battery and a 3-axis iMEMS accelerometer (ADXL345 Analog Devices) set to record 11-bit digital measurement of acceleration with a range of ± 4 g and a sampling rate of 50 samples per second using a digital micro-controller and data storage on flash memory. The logger can be programmed to delivering vibrations that remind subjects to take their PD medications. Consumption of medications is acknowledged by swiping the logger’s smart screen. The logger also has sensors to detect whether the device is being worn. The device is water resistant. The logger has been designed and approved for easy cleaning and reuse and after 6 days of recording, data are uploaded to the cloud for application of the algorithms.

#### The Algorithms

An expert system approach was used to model neurologists’ recognition of bradykinesia and dyskinesia on accelerometry data. Inputs to the expert system included Mean Spectral Power (MSP) within bands of acceleration between 0.2 and 4 Hz, peak acceleration, and the amount of time within these epochs that there was no movement. These inputs were weighted to model neurologists’ rating of bradykinesia and dyskinesia and to produce a bradykinesia score (BKS) and dyskinesia score (DKS) every 2 min.

The PKG is the graphical representation of the BKS and DKS collected every 2 min over an extended period (typically 6 days). As the device is worn at night to obtain a sleep score, these were presented in the PKG, as well as scores of day time sleepiness and inactivity [22]. Other scores included compliance with the reminders, tremor scores [23], and times when the logger was not worn. Over the 6 days of continuous recording there were 4320 2-min data points. The PKG plotted the mean BKS and DKS (with a smoothing function) against the time of day. The time that medications were due and consumed were also shown, making it possible to assess whether there were dose related variations in BKS or DKS and how the median value at any time of day compared with a normal subject. There were also raster plots showing the time when tremor or sleep occurred or when the logger was not on the wrist. Finally, numerical scores for percent of the time in tremor or asleep and BKS and DKS scores were presented. The numerical output relevant to this study is summarized further in the Glossary that follows. The reader is referred to the company’s web site for further details (http://www.globalkineticscorporation.com.au/) and to other publications [21,22,23,24,25,26,27,28,29,30]. 

### 2.2. Glossary of PKG Terms

The following scores were calculated from data obtained between 09:00 and 18:00 and excludes periods when the logger was not worn or when the subject was immobile (see PTI below).
Median BKS. The median BKS was the 50th percentile of the BKS for all days that the PKG was worn (usually 6 days).Interquartile range of BKS. This was the interquartile range of the BKS of all epochs used to calculate the median BKS and was a measure of fluctuation of the BKS [29].Percent time in bradykinesia (PTB). Epochs whose BKS lie between 26.1 and 49.4 and whose 25th percentile of the BKS > 18.5 and the 90th percentile of BKS < 80. As well, any epoch whose BKS > 49.9 but contained tremor was included. These epochs were passed through a moving median filter and most of the epochs in a window of 30 min must be >26 for the center to be “off”.Median DKS: The median DKS was the 50th percentile for all the days that the PKG was worn [21]. Brisk walking introduced resonant peaks in the frequency of acceleration relevant to the DKS algorithm and thus artefactually increased the DKS. An algorithm was used to detect and remove epochs affected in this way.Interquartile range of DKS: This was the interquartile range of the DKS of all epochs used to calculate the median BKS and is a measure of fluctuation of the DKS [29].Percent Time in Dyskinesia (PTD): Those DKS used to estimate the median DKS were passed through a median filter (most of the epochs in the filer period must be in the dyskinetic range (DKS > 7) for the center to be classed as dyskinetic).The percent time with tremor (PTT): This was the percent of 2 min epochs estimated over all days that the PKG was worn that contained tremor [23].The percent time immobile (PTI): This was the percent of 2 min epochs with BKS > 80 from all days that the PKG was worn. These scores were associated with daytime sleep [22].Doses of levodopa/day. This was calculated from the number of reminders programmed into the logger.

## 3. Results

### 3.1. Development of DAT Classifier Scores

The first objective was to create a classifier that accurately reflects the separation of PwP by clinicians into those who should be referred for DAT and those who are not yet ready and can still be managed by oral medications. The broad processes for building such a classifier was outlined in the Introduction. What follows is a more detailed description of the steps taken.

#### 3.1.1. Clinical Sorting

DAT should be considered when there are troublesome “off” periods or dyskinesia that cannot be addressed by the optimal deployment of oral therapies [16]. Thus, deciding that someone is suitable for DAT requires three steps: The first is identifying PwP who met the first criterion of DAT suitability by having excessive periods of bradykinesia and/or dyskinesia: The second criterion is to recognize that this bradykinesia and/or dyskinesia could not be reduced by manipulating oral therapies; the third criterion is to identify various (usually non-motor) contraindications that might deter from proceeding to DAT and this is usually a task carried out in specialist centers. Failure to recognize the first two criteria are the main reasons for delayed or inappropriate referrals for DAT. Our approach here was to build a classifier that recognized PwP who met the first criterion AND that changed classification according to whether or not the second criterion was met. That is, classification was changed from DAT suitability to be no longer suitable because manipulation of oral therapies had addressed the excessive periods of bradykinesia and/or dyskinesia. 

Four clinicians sorted 172 PwP according to whether or not they met both criterion 1 AND 2 (above) and were thus suitable for DAT on motor grounds (Table 1). They were thus grouped as clinical “criteria positive (CP)” or clinical “criteria negative” (CN). Clinicians were asked explicitly not to consider non-motor factors in this decision. Because pump assisted therapies are included as DAT and were readily available at these centers, many of the more stringent non motor contraindications required for DBS were less relevant to DAT as a whole. Nevertheless, we suspect that motor criteria were less rigidly applied in younger subjects than in those where age or other non-motor factors were present. A little less than 2/3 of the PwP classified as CP did proceed to DAT. Most subjects who did not proceed to DAT were older and had declined DAT although it was recommended. CP subjects were younger and had worse scores as measured by the UPDRS II, IV, and total UPDRS. While UPDRS III scores of the two sets were not significantly different, UPDRS I score were worse in CN PwP. It is relevant that the UPDRS IV and the PKG’s median DKS and PTD were worse in the CP group, whereas the PKG’s median BKS and PTB were worse in the CN group (Table 1).

#### 3.1.2. Training and Re-Test Set Composition

Ideally, for classifier generation, the training and re-test sets should be randomly selected from the same integral set of PwP: They would be sets with similar distributions that completely represented the population of PwP from whom DAT candidates would be selected. The 172 PwP were randomly assigned to a construction set (N = 112) and a test set (N = 60, Appendix A). Note that the “construction set” refers to the set in which all the processes required to fully develop a classifier, including training, developing, cross-validating, and performance evaluation were carried out. The two were statistically similar sets with respect to clinical and PKG scores (*p*-values > 0.05, non-parametric *t*-test). The CP and CN classes from one set were compared with the matching group in the other set and found to be statistically similar (*p*-values > 0.05, non-parametric *t*-test, Appendix A). 

#### 3.1.3. Parameter Selection

The choice and refinement of input features is a key process in building classifiers. While there are numerous approaches to extracting features, we chose as a first step, to select intuitively PKG variables based on information from the literature (see Reference [16] for a review) and from experience gained in developing a previous classifier [19]. The literature (see reference [16] for a review) indicates that in identifying subjects suitable for DAT, clinicians establish whether there are excessive periods of bradykinesia or dyskinesia that cannot be reduced by manipulating oral therapies. They use patient estimates of time “off”, PTD and frequency of dosing to understand this. The PKG representation of this information included the median BKS and DKS and their interquartile ranges, PTB and PTB, PTT, age and number of reminders (see the Methods section for a full description of these terms). 

The next step was to refine the selection of predictor variables (that are pre-selected based on clinical relevance) using statistical evidence. We use a supervised model independent method for identifying any type of correlation (linear or non-linear) between these PKG variables and the labels of CP or CN, referred to as the joint mutual information maximization (JMIM) [31], to calculate the PKG variables and the CP and CN labels in the construction set. The PKG variable with the maximum mutual information with CP and CN was selected as the “first variable” and the process was repeated to iteratively add the variable that maximizes relevancy to redundancy relationships to CP and CN, given the subset of the already selected PKG variables (up to the previous iteration). The joint mutual information [31] of these PKG variables with CP and CN labels are shown in Table 2, which shows that the set of intuitively selected PKG variables all contain non-redundant relevant information about the CP and CN classes (Figure 1). Age was also not present because it did not contain significant joint mutual information with the CP and CN classes. Most likely this is because the clinical categorization was on motor grounds and clinicians were asked to disregard age or non-motor factors. The absence of age in the parameters is reviewed further in the Discussion. It should be noted that while principle component analyses (PCA) was performed to analyse the feature space and model selection, it was not used as a dimensionality reduction step in the model, which uses all 8 dimensions which are reasonably small with respect to the size of the training set. The linear and non-linear relationship of the feature space with the CP and CN classes is captured in the classifier model. 

#### 3.1.4. Model Building

The next step was to build statistical models using these parameters to predict the clinical classification of CP and CN in the construction set. The process commenced with a statistical model using the parameters in Table 2 and a process of k-fold cross validation. K-fold cross validation randomly divides the construction set into k subsets (folds), with equal-sizes [32]. The first fold is used as a validation set, and the remaining k-1 folds are used as the training set. The statistical model was applied to these sub-samples and their capacity to fit the clinical classification of CP or CN. The objective was to optimize the area under the receiver operating characteristic (ROC) curve through an iterative process of parameter addition and elimination. The set of parameters in Table 2 were all identified as contributing to improved performance of the classification models described below. 

#### 3.1.5. Outputs of Statistical Model

Several different models were considered and were subject to a preliminary examination. Two were fully developed: A linear and a non-linear model. The linear model used Logistic Regression to sort subjects in the construction set as either CP and CN using a weighted sum of the above scores and a bias term. The weight and bias terms were obtained by minimizing the prediction error for the training set. The logistic function was then applied to this weighted sum to yield a probabilistic score between 0 and 100, which provides a simple yet effective algorithm for sorting subjects as either CP and CN. 

The non-linear model was developed using the support vector classifier (SVC) [33]. This tries to separate two classes with the largest margin and, thus, was examined as a means of separating the CP and CN classes. Although SVC has been shown to perform better than linear discriminant analysis in terms of class separation, we did confirm that the nonlinear model was superior to the linear model on the original feature space (data not presented here). A nonlinear kernel for SVC was used for several reasons. First, it was apparent at cross validation that logistic regression relative underfitted the construction set compared to the SVC, suggesting that a nonlinear model should be employed. Second, the nonlinear nature of some PKG variables (such as DKS) implies that nonlinear models on the feature space are called for. Third, while principle component analysis shows that the first three components explain most of the variance in the feature space (Figure 1A), it is evident that a linear hyperplane on the three-dimensional space of the first three components could not separate the two classes of CP and CN, whereas a bell-shaped surface would (Figure 1A,B). While a linear separation was unlikely to be achieved by adding more dimension from the set of available components, a kernel such as gaussian radial basis function could make this transformation possible. The kernel SVC maps the features into a higher dimensional space where a hyperplane can separate the two classes. Thus, in the transformed feature space, the classifier is linear. 

While this examination of the original feature space led to the choice of a nonlinear classifier, it also came with a higher chance of generalization error. This was addressed with an ensemble learning approach. Acceptable bias-variance trade-off can be achieved by avoiding an over-complicated model but also by parameter selection (described above) when dealing with small training sets. An ensemble learning method referred to as bootstrap aggregating (or bagging), which randomly draws samples from the training set with replacement, trains the base classifier using each randomly drawn subset separately and then averages their individual predictions [34]. Then, it declares the majority vote as the final decision. L2 regularization was also performed to further reduce the chance of over-fitting on base classifiers [35]. 

Using the above approach, multiple linear and SVC models were built and trained with a randomly selected subset of the construction set. The predicted DAT classifier was then obtained by aggregating the predictions of all the individual classifiers. The hyperparameters of the base and bagging classifiers were tuned to obtain the optimal area under the curve (when performing multiple fold cross-validating on the construction set).

#### 3.1.6. Outputs of the Classifiers

The performance of the optimized non-linear classifier in matching the Movement Disorder clinician’s decisions of whether a subject in the construction set does or does not meet the clinical motor criteria for DAT is shown in Figure 2a,b and in Table 3. While some PwP clearly do not require DAT and there are others who are unequivocally ready, there is some uncertainty about the classification of those PwP who are in transition from one category to the other. Thus, the classifier was designed to provide a value, which increased from 0 to 100, reflecting a greater likelihood of the need for DAT. By deploying a characteristic of the SVC (on which it was based), this score separates the two clinical classifications (CP and CN) with the largest margin. The effect is to push PwP with low and high-risk towards the extreme ends of the score (as in Figure 2c). 

### 3.2. Performance of the DAT Classifier Scores on the Test Set

Once the non-linear classifier was optimized on the construction set, it was then tested on the test set where ideally, it should have only slightly inferior performance, which was indeed the case (Figure 2a,b and Table 3). Finally, the non-linear classifier was run on the whole data set (construction and test sets combined) with similar outcomes to those on the test set (Figure 2c,d and Table 3). Its performance was also compared with the linear score. The non-linear classifier has the advantage of classifying patients into low (score < 20), intermediate (20–60) and high risk (≥60) ranges. As the best performing classifier, we will refer to the output of the non-linear classifier as the DAT classifier score and use it in the further validations described below. 

### 3.3. Clinical Performance of the DAT Classifier Score

A central reason for this study was to produce a screening tool to aid the non-specialist in making timely referrals for DAT, without burdening these centers with too many unsuitable cases [5,10]. As discussed above, two criteria for timely referral are the presence of; (a) excessive periods of bradykinesia and/or dyskinesia; (b) that cannot be reduced by manipulating oral therapies (i.e., the second criterion). Our aim was to build a classifier that recognized PwP who met the first criteria AND which changed its DAT classifier score to reflect reductions in bradykinesia and/or dyskinesia brought about by manipulation of oral therapies. In other words, PwP whose levels of bradykinesia and/or dyskinesia warrant consideration of DAT should have a commensurately high DAT classifier score but where a change in oral therapies resulted in clinical improvement, then the DAT classifier score should also fall, reflecting this new state. Otherwise, the DAT classifier score is not performing in a way that will help a referring clinician. 

Although the ROC on the test set provided a strong validation, it is important to understand why the DAT classifier score misclassified (compared to the clinician’s decision) individual cases. While these may be relatively few, understanding the reasons for their presence will at the least provide advice on the limitations and pitfalls in using the DAT classifier score but may also lead to insights into how to improve it and the PKG. Some reasons for misclassification might include: (i) The PKG variables are affected by artefacts or were erroneous (e.g., failed to detect lower limb dyskinesia); (ii) some but not all DAT specialist may have believed DAT was indicated and thus the clinical classification is uncertain; (iii) the algorithm can be improved. From an algorithm designer’s point of view, it is important to understand those errors that arise from the PKG or the algorithm and to advise potential users of them. Having built the classifier, the performance of the classifier before and after a change in therapies was examined. First, sorces before and after DBS were compared: The assumption was that following a DAT the scores should improve to the point of indicating that the factors that triggered the referral have now resolved. We also compared the DAT classifier scores before and after oral therapies.

#### 3.3.1. Testing the DAT Classifier Score on PwP Already Selected for DBS

In five clinics in Australia, 31 of the 172 subjects had undergone DBS and had PKGs before and 6 months after commencement of stimulation. Prior to surgery, the DAT classifier score was above the threshold, indicating DAT suitability in all but four cases (green circles in Figure 3a). When asked about these four subjects, the managing clinicians advised that these cases were at the low end of the criteria for receiving DBS, mainly because of age. It is relevant that of the three main sites contributing subjects to this study, the mean DAT classifier score of two sites was approximately 90, whereas the mean DAT classifier score of the third site, which included most of the subjects marked by the green circles was 65. There may be many reasons for this variation but the threshold of criteria for surgery may well be an important factor. 

In broad terms, the change in the DAT classifier score following surgery could be predicted from the pre-surgery score: Those with the highest score had the greatest reduction following surgery (Figure 3b). The four cases with lower DAT classifier scores did not improve greatly and this was reflected in clinical notes and their UPDRS scores. There were seven cases (red circles), whose improvement in DAT classifier score was not as great as that achieved by the other cases. In these cases, there was ongoing dyskinesia and frequent consumption of levodopa (5 or more doses/day). As well, in two of these cases, it is possible that exercise may have artifactually elevated the dyskinesia score. When the change in DAT classifier score from before DBS to after DBS (Δ DAT classifier score, X axis Figure 3b) was plotted against the pre-treatment DAT classifier score (Y axis Figure 3b) it was apparent that most subject lie along a line shown as a grey shaded area in Figure 3b. The seven cases (red circles), whose DAT classifier score did not improve, fell outside this region. 

In summary, the DAT classifier scores predicted that 87% were ready for DAT on the pre-surgery PKG and there was a substantial improvement in their scores post-DBS. The remaining 13% were cases whose scores were considered by their clinicians to be at the threshold for requiring DBS. In keeping with this, there was little improvement in their DAT classifier scores. 

#### 3.3.2. Testing the Performance of the DAT Classifier Score in Identifying Potential DAT Candidates in Typical Clinic Population of PD

The purpose of developing a DAT classifier score was to aid non-specialist clinicians in selecting subjects for DAT referrals. To assess its performance in this regard, we examined the DAT classifier score in 81 PwP who were representative of the spectrum of PwP managed in a non-specialized Movement Disorder center [26]. These PwP were usually managed by a geriatrician led Parkinson’s Disease service who did not themselves initiate DAT, and was thus typical of the service that might benefit from such a tool. They were part of another study [26] which required a single Movement Disorder specialist to classify cases according to whether they have motor symptoms of PD that did not require treatment (optimized); required a change in oral therapies; were referred for DAT; had contraindications that prevented motor symptoms from being adequately treated (Figure 3c,d). Note that the DAT classifier score was not available to the clinician when making these classifications. Analyses of the DAT classifier score of these 81 PwP allowed the following questions to be addressed:Did the DAT classifier score identify all the subjects that the Movement Disorder specialist referred for DAT?How did the DAT classifier score change in those the Movement Disorder specialist regarded as requiring a change in oral therapies?

The median and 75th percentile of DAT classifier score of PwP whose symptoms were optimally controlled according to the Movement Disorder specialist (Figure 3c), were 5 and 35 respectively (i.e., low). The score of two cases was in the high-risk range (>60) and factors that may have elevated the score in these two subjects, were dyskinesia and artefactual influence of exercise. The question of the accuracy of the Movement Disorder specialist’s classification and the contribution of the artefact is further addressed in the Discussion. For comparison, the DAT classifier score was obtained for 192 subjects without PD, aged over 60: Their median DAT classifier score was zero and the 90th percentile was 2.25 (Figure 2c).

Undertreated bradykinesia was the most common reason treatment was changed in this population (“Pre Oral R” in Figure 3c). Prior to changing oral therapies, 49% (n = 19) of “Pre Oral R” subjects in Figure 3c had DAT classifier scores above 60, and 63% (n = 12) of these were referred for DAT by the study physician after attempts at optimization (Figure 3c: DAT ready). DAT classifier scores fell below 60 after treatment change in only one of these cases (DAT R in Figure 3c). In many of these cases, treatment changes were regarded as worthwhile by PwP (data not shown) but the clinician’s view was that DAT would achieve further improvements in fluctuations, off time, and dyskinesia without the burden of a high number of dose/day. The DAT classifier score remained high, reflecting the clinicians view. 

Of the remaining “Pre Oral R” subjects, the DAT classifier score fell below 60 in 85% and below 40 in 70% (Post Oral R in Figure 3c). The DAT classifier scores may have remained high in the two remaining cases (represented as blue circles in Figure 3c) because one case with dementia did not respond to levodopa and in the other case, the dyskinesia scores may have been artifactually elevated from exercise. Thus, one of these cases could be more appropriately placed in the group where a change in therapy was contraindicated (“Cont.” column in Figure 3c). Finally, there were three cases classified as “Pre Oral R” (mainly because of bradykinesia), whose DAT classifier scores before treatment were low risk but increased markedly after treatment and shifted into the high-risk range (green circles). The DAT classifier score increased because the increasing treatment (and resolving bradykinesia) made these cases fluctuate. Reflecting this, the Movement Disorder specialist identified these cases as ready for DAT after attempting to optimize therapy. 

These findings can be summarized by plotting the difference between DAT classifier score before and after treatment change (Δ DAT classifier score, X axis Figure 3d) against the pre-treatment DAT classifier score (Y axis Figure 3d). This suggests that those subjects whose treatment can be optimized without resorting to DAT (black circles) will lie in the predicted response range (grey region of Figure 3d). Cases that do not lie in this region are either those that did not respond to oral therapy and require DAT (red circles) or those whose need for DAT was unmasked by increasing oral therapies (green circles). 

Combining and plotting the DBS data (Figure 3b) and the population data (Figure 3d) leads us to the conclusion that the DAT classifier score is performing as expected.
It predicted those requiring DAT with high test sensitivity.When a change in therapy (oral or DBS) in PwP with a high DAT classifier score led to an improvement, the post-treatment score fell predictably. This is shown as the region of grey shading of Figure 3e. The better the control achieved by the change in therapy, the greater the reduction in the score along this grey zone.When a change of therapy (oral or DBS) in PwP with a high DAT classifier score failed to result in optimization, the score did not fall and appeared in the pink region in Figure 3e.When a change in therapy addressed undertreatment but uncovered fluctuations and or dyskinesia the DAT score lay in the green shaded area (in Figure 3). If the change resulted in a high DAT score, it usually revealed a need for DAT.Further work is required to better understand artefacts in input data (from the PKG).

Overall this suggests that applying a DAT classifier score to the typical population of PwP managed outside the specialist clinic will assist the clinician in deciding whether or not to refer for DAT. In essence, a low DAT classifier score does not need a referral for consideration of DAT and a high DAT classifier score *which cannot be corrected by optimizing oral therapy* should be referred for expert consideration for DAT. This is in keeping with clinician practice where the criteria for DAT are excessive periods of bradykinesia or dyskinesia that cannot be reduced by manipulating oral therapies. If these findings were to hold up in a larger study, it suggests that there would be few false negatives (i.e., subjects who might benefit from DAT who are overlooked) and less than 20% false positive (i.e., unnecessary referrals for consideration of DAT).

#### 3.3.3. Testing the DAT Classifier Score on PwP Over an Extended Follow-Up from Diagnosis

As a further test of the clinical validity of the performance of the DAT classifier score, we examined serial DAT classifier scores obtained from 22 subjects who were referred to a Movement Disorder Clinic at or near the time of diagnosis of PD. The clinical care was provided by a single clinician and PKGs were available from the presentation and then at least annually (Figure 4) for a median of 5 years (min 4 years, max 7 years). Over that time, the clinician considered that eight had developed the clinical criteria for DAT; five had developed fluctuations and would soon meet the clinical criteria for DAT; nine were not ready for DAT. The distinction between “meeting the criteria” and “soon to meet the criteria” was simply that in the former, the clinician had already discussed DAT with the PwP whereas, in the later, the clinician had noted that the need for DAT was imminent, they had not yet raised it as possibility with the patient. Those in whom DAT was already indicated had DAT classifier scores in the high-risk range, which were very similar to those who would soon meet the criteria. Subjects that did not meet the criteria had low DAT classifier scores. It is interesting that the DAT classifier scores had reached medium risk levels by 24 months in those in whom DAT was already indicated, which was about a year sooner than in those who would meet the criteria soon. The DAT classifier score remained low (<20) in those who did not require DAT, albeit with modest variability. This lends support to the possibility of monitoring subjects with a DAT classifier score to improve the optimization of oral therapy but also to ensure timely referral for DAT. 

## 4. Discussion

The aim of this study was to use data from an accelerometry based measurement system to build a DAT classifier score that gave the likelihood of a PwP meeting the motor criteria for DAT (bearing in mind that the two criteria for timely referral are recognizing that (a) excessive periods of bradykinesia and/or dyskinesia are present (b) which cannot be reduced by manipulating oral therapies). The main findings of the study are:The information provided by six days of accelerometry recordings from the wrist provides sufficient information to build a classifier that distinguished with high sensitivity and specificity between those subjects that did or did not meet the first clinical criterion for DAT, according to classification by specialist clinicians.The DAT classifier score correctly assigned subjects to DBS in cases preselected for surgery in 87% cases and the remaining miss-assigned cases may well have provoked discussion amongst specialist about whether they were indeed suitable or not.Cases where excessive periods of bradykinesia or dyskinesia could not be corrected by oral therapy were recognized as subjects whose DAT classifier scores remained high following changes in oral therapy directed at optimizing treatment. Thus a clinician using the DAT classifier scores score would have correctly identified these subjects as eligible for DAT.Serially following the change in DAT classifier scores of newly diagnosed PwP correctly identified subjects as they developed the clinical criteria for DAT.The change in DAT classifier scores following therapeutic intervention has led us to propose that if the change in the score does not follow a standard response then it indicates either the need for DAT or failure in its administration.

Each of these points, however, raises further discussion points which are addressed below. 

### 4.1. How Can Accelerometry Data Recording from One Wrist Provide Enough Information for Identifying Subjects Who Might Benefit from DAT

Suitable candidates for DAT are recognized by the presence of increased off-time and/or dyskinesia in subjects taking five or more dose/day [16] that cannot be correct by further manipulation of oral therapies. The variables provided by PKG are objective measures of these same factors that are considered clinically and the previous pilot study gave some insights as to the weighting of these variables used by experienced clinicians [19]. While there are many other factors taken into account before DAT is recommended, motor criteria form the main basis for inclusion and referral by the non-expert. 

Recording of acceleration from the wrist can overlook motor disturbance of the head, trunk or lower limbs that are not also present on the wrist wearing the logger. While this occurs in up to 10% of cases, it rarely influences the score [26]. As well, exercise can artifactually elevate the dyskinesia score and thus the DAT classifier score, as may have occurred in one case in Figure 3c. The number of doses/day is also an important driver of the DAT classifier score (Table 2) and thus a frequent number of reminders for reasons other than a short duration of benefit from a dose of levodopa may also artifactual inflate the score. 

Age of the participant was included in the original parameter set but was excluded because it failed to contribute significant mutual information (Table 2). At first, this might seem surprising, because age does influence the choice of people for DBS, however, it is less critical in the non-DBS forms of DAT. Our interest was to build a classifier that selected subjects as suitable for DAT on motor grounds as this is the main inclusion criteria for DAT and allows expert centers to then exclude subjects for the various non-motor reasons including age. Clinicians were asked to focus only on motor criteria and disregard age, cognitive factors, and other non-motor aspects in making their classification, and the fact that age did not have a strong correlation with the labels of CP or CN suggest that, in the main, they did this. However, we cannot exclude the possibility that some non-motor factors including age did not influence the stringency with which they chose subjects as “ready” for DAT. 

### 4.2. The Performance of the Classifier

The classifiers were modelled on and tested against classifications made by clinicians. The implied assumption is that the standard on which the classifier is modelled is always correct, uniform and univariable. However clinical experience tells us this is unlikely and the variability in scores in the three centers referring for DBS, suggests variation in practice. Thus, one of the reasons that the sensitivity, specificity, and AUC of the ROC against the test set were not higher may be clinician inconsistency (between and within). As well error will arise from the accuracy of the PKG variables. As described above the PKG’s dyskinesia scores may underestimated dyskinesia when the trunk or lower limbs are primarily affected and overestimate when there is prolonged exercise at a specific time each day. The number of cases where this may be a factor is probably <5%. 

### 4.3. Does the DAT Classifier Score Sort According to Its Purpose

A PwP will slowly and progressively develop the clinical criteria for DAT over months to years as depicted in Figure 4. However, the exact point at which the criteria are met is not clear cut, thus the DAT classifier score was designed to give the likelihood or “risk” (in the statistical sense) of requiring DAT. In the spirit of this, we are reluctant to provide a “cut-off” score, as it leads to a hard transition rather than progressively increasing the likelihood of needing DAT as the score increases. The cut-off or optimum score is shown in Table 3, which provided the best sensitivity and specificity, but as discussed elsewhere [36], the boundary should be chosen for the purpose at hand. Thus, it may be that lowering the optimum score may reduce false negatives, which may be acceptable for use as a referral tool. If the threshold for referral was lowered from the optimum score (61) to say 45, false negatives are reduced by half but the number of unnecessary referrals is increased. Further studies might assist in understanding the practical criteria for use in a particular clinical setting. In deciding an acceptable false negative rate, it is important to note that the comparison should be with current rates that currently occur for a late or absent referral. There are few studies that address this, but in the recent study of Tasmania, from which we drew data for Figure 3c, 19% were considered as suitable for DAT but had not been referred by the local clinic. 

The data presented in Figure 3e appears to confirm that the DAT classifier score behaved appropriately in a clinical setting. It suggests that the score could be useful in practice when a clinician wishes to first assess whether modifying oral therapy might be affective before referring. If the score fell along the line of the predicted response, then it suggests that the need for DAT has been postponed or possibly avoided. On the other hand, if it remained high and moved to the left on Figure 3e and into the range where the indication for DAT persisted, then a referral would be indicated. Clearly, a future study would be needed to address whether referrals emanating from assistance to the clinician from the DAT classifier score resulted in a high proportion of relevant referrals while not overlooking falsely excluded cases. 

## 5. Conclusions

PD initially presents with bradykinesia, which is relatively simple to treat in the first few years. After that time, management becomes challenging as the fluctuations between bradykinesia and dyskinesia with each dose and shortening of dose effect to approximately three hours. While DAT is among the most effective means for managing this stage of PD, many PwP for whom this therapy would be appropriate, miss out because their managing clinician fails to recognize the indications. The PKG system was used because it provides objective measures of severity of bradykinesia and dyskinesia, time “off”, PTD, and frequency of dosing, which are the same measures that clinicians extract by history to establish whether there are excessive periods of bradykinesia or dyskinesia that cannot be reduced by manipulating oral therapies. Thus, it was likely that it would provide input features that could be used to build a DAT classifier score of the likelihood that a PwP meets the motor indications for DAT as determined by clinical classification. The main findings of the study were as follows.
The information from the PKG could be used to build a classifier that identified with high sensitivity and specificity, PwP who specialist clinicians had identified as meeting the criteria for DAT from those that did not. Thus, the DAT classifier score was successful in identifying PwP who met the first criterion for DAT suitability: Having excessive periods of bradykinesia and/or dyskinesia achieves.The DAT classifier score correctly assigned subjects to DBS in 87% of cases who had already been preselected for surgery. The remaining miss-assigned cases may not have been considered by all specialists as suitable cases. This is in keeping with the current discussion in the movement disorder specialty around how early in the disease DBS is a suitable therapy.Cases where excessive periods of bradykinesia or dyskinesia could not be corrected by oral therapy were PwP who met the second criterion for DAT: That is, excessive periods of bradykinesia and/or dyskinesia could not be reduced by manipulating oral therapies. The DAT classifier met this second criterion because the scores remained high despite efforts in using oral therapy to optimize treatment. Thus, a clinician using the DAT classifier scores score would have correctly identified these subjects as eligible for DAT.Using an effective DAT classifier score to measure PwP from diagnosis to the onset of excessive periods of bradykinesia or dyskinesia that could not be corrected by oral therapy, which should see a commensurate change in the DAT classifier score as a movement disorder specialist was more likely to consider introducing DAT. The DAT classifier performed well under these circumstances.The change in DAT classifier scores following a therapeutic intervention has led us to propose that that there is a predictable change in the score if the intervention is successful. Figure 3e might suggest that the response could be Δ DAT classifier scores (before-after intervention) = DAT classifier score (before intervention) − 20. A response that does not follow this pattern indicates either the need for DAT or failure in therapeutic administration.

Further studies are required to establish whether this optimism is justified. The DAT classifier score described here has been modelled on the clinical decisions of a relatively small number of clinicians operating out of only a few clinics in one country. However, it is important to understand the aim is not to model the behavior of the average clinician but to model clinical behavior that results in good outcomes for PwP.

## Figures and Tables

**Figure 1 sensors-19-02241-f001:**
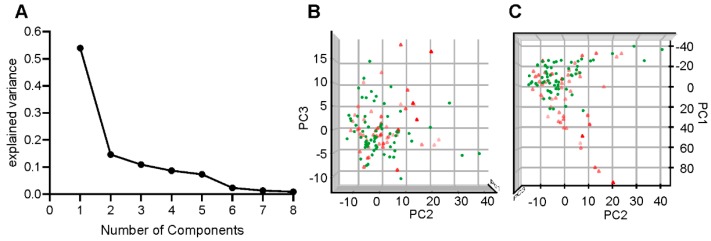
(**A**) Is a plot of the first eight components (Y axis) of a PCA of the PKG’s parameters and the classification into CP and CN against the explained variance (X axis). (**B**,**C**) are three-dimensional plots of the first three components of the PCA. Points shown by green circles represent cases classified as CN, whereas red diamonds represent cases classified as CP (color intensity indicate position on the Z axis).

**Figure 2 sensors-19-02241-f002:**
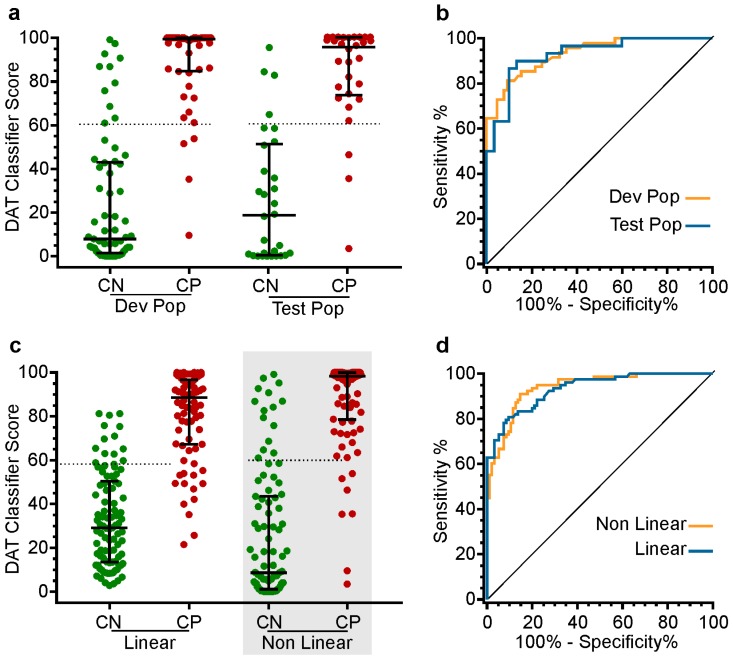
(**a**) Shows the scores from the non-linear classifier as applied to the construction set (left pair) and the test set (right pair). Green circles represent PwP cases clinically classed as CN, whereas the red circles represent PwP cases clinically classed as CP. (**b**) Shows the outputs of the receiver operator statistic applied to the scores in Figure 2a. The orange curve shows the construction set and the blue line shows the test population. (**c**) Shows the scores from the linear and non-linear classifier when applied to the whole set. The grey shading around the non-linear classifier is to indicate that it was renamed the DAT classifier Score. (**d**) Shows the receiver operating characteristic (ROC) curve applied to the two groups of data shown in Figure 2c.

**Figure 3 sensors-19-02241-f003:**
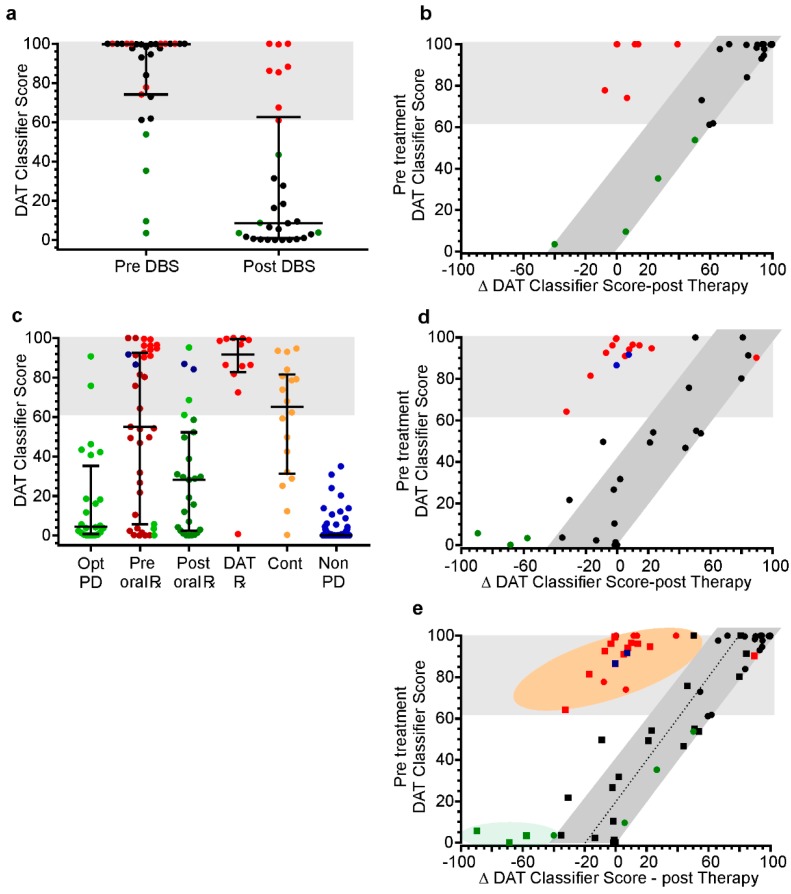
(**a**) Compares DAT classifier scores obtained from PwP prior to receiving DBS and six months after DBS and scores in the shaded area indicate a high risk of DAT being indicated. Green circles indicate four cases (green circles) who were not in this zone. Red circles represent cases whose scores did not fall below the high risk region after DBS. (**b**) Is a plot of each individual PwP’s DAT classifier score prior to DBS (Y axis) plotted against the change in DAT classifier scores after DBS (X axis). The coloring of the circles represents the same cases as in Figure 2a. The grey shading represents the predicted response range. (**c**) Compares DAT classifier scores from the PwP representing all PwP in a single population, (1) Opt PD (light green circles) were optimally controlled cases; (2) Pre Oral R shows the DAT classifier scores of PwP prior to attempting a change in oral therapy. The green circles indicate PwP who were under-treated, but treatment resulted in an increase in DAT classifier scores and their treating clinician now thought DAT was indicated. The blue circles represent two PwP whose DAT classifier scores remained high because of a lack of responsiveness to levodopa and artifactually elevated dyskinesia; (3) Post Oral R are cases whose DAT classifier scores were reduced when measured after the change in therapy (shown as dark red circles in Pre Oral R); (4) DAT R shows cases referred for DAT following attempts to change oral therapy (light red circles Pre Oral R). (**d**) Shows the individual PwP’s (pre Oral R) DAT classifier scores prior to a change in oral therapy (Y axis) plotted against the change in DAT classifier scores following that change (X axis). The coloring of the circles represents the same cases as in Figure 3c. (**e**) Combines Figure 3b,d to propose a response region (grey shading), where an effective optimization of therapy will fall. The region shade orange indicates failed optimization and the region in green indicates where increasing therapy has led an increased DAT classifier score.

**Figure 4 sensors-19-02241-f004:**
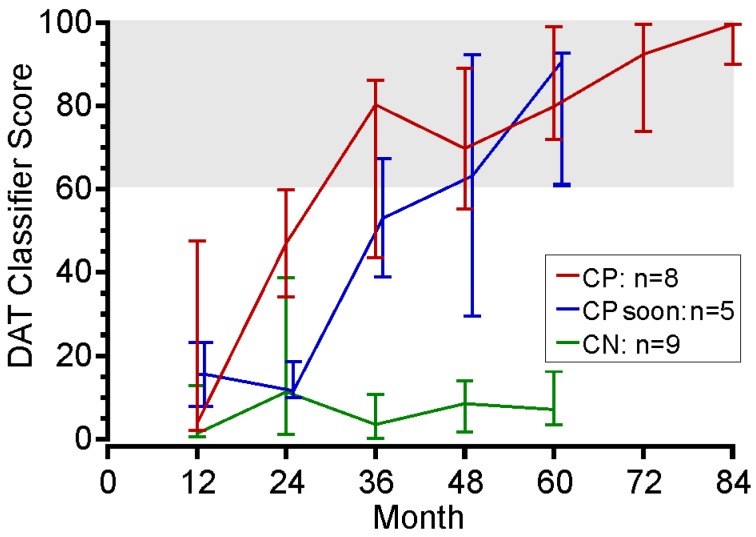
A plot of the mean DAT classifier scores (highest and lowest) for subjects who met the clinical criteria for DAT (CP, red line), would soon meet the criteria (CP-soon, blue line) or did not yet meet the criteria (CN, green line). Note that this assessment was applied at the PwP’s most recent visit and that not all subjects have been followed for the same length of time.

**Table 1 sensors-19-02241-t001:** Comparisons of the clinical and Parkinson’s KinetiGraph (PKG) characteristics of people with Parkinson’s (PwP) who were classified according to whether they met the clinical criteria for device-assisted therapies (DAT) (criteria positive (CP)) or did not (criteria negative (CN)) for DAT.

N Total = 172	CP	CN	Δ	*p* Value
Male	66%	70%		
Female	34%	30%		
Age	62 (56–67)	71 (66–75)	9	0.13
UPDRS I	6 (3–10.5)	8 (5–13)	−2	0.02
UPDRS II	13 (10–18)	7 (4–12)	6	0.0001
UPDRS III	27 (19–37)	25 (18–35)	2	0.3
UPDRS IV	7 (4–9)	1 (0–4)	6	0.0001
UPDRS Total	54 (46–72)	43 (32–59)	11	0.001
Median BKS	20.8 (16.4–25.4)	24 (21.7–27)	−3.2	0.0001
PTB	37.5 (19.1–55.9)	47.7 (35–65.1)	10.2% (1.6 h)	0.0004
Median DKS	5.1 (2.4–13.5)	2 (0.9–3.8)	3.1	0.0001
PTD	23.9 (10.2–46.7)	7.3 (2.9–13.3)	16.6% (2.7 h)	0.0001
DBSS	0.96 (0.82–0.99)	0.16 (0.02–0.5)	0.8	0.0001
doses/day	5 (5–6)	4 (3–4)	1	0.0001
PTT	0.8 (0.3–2.1)	0.8 (0.4–2)	0	0.6
PTI	4.1 (1.5–.8)	4.8 (2.8–8.9)	0.7	0.02

**Table 2 sensors-19-02241-t002:** The incremental joint mutual information of PKG variables with CP and CN labels.

PKG Variable	Joint Mutual Information	Clinical Information Represented by the PKG Data
Doses of l-dopa	0.25	Dose of levodopa/day
DKS 75	0.11	Severity of dyskinesia
BKS 25	0.08	Severity of bradykinesia
BKS 75	0.08	Fluctuation in bradykinesia
PT in DK	0.06	Time in “troublesome” dyskinesia
PTI	0.05	Time asleep during the day
PTT	0.04	Time with tremor
PTO	0.03	Time “off”

**Table 3 sensors-19-02241-t003:** The performance of the optimized non-linear classifier in matching the Movement Disorder clinician’s decisions of whether a subject in the construction set met the clinical criteria for DAT (criteria positive (CP)) or did not (criteria negative (CN)).

	Construction Set		Test Set		Full Data Set	
	Linear	Non-Linear	Linear	Non-Linear	Linear	Non-Linear
Area	0.92	0.94	0.93	0.93	0.93	0.93
Optimum score	58.3	61.1	58.3	61.1	58.3	61.1
Sensitivity	85.5	91.7	80	89	80	89
Specificity	84.6	84.6	88.5	86.6	88.5	86.6

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
