# Peer review of "The Use of Data from the Parkinson’s KinetiGraph to Identify Potential Candidates for Device Assisted Therapies"

_sensors, 2019, doi:10.3390/s19102241_

Round 1
Reviewer 1 Report
1. Motivation of your work is too short. Please consider adding the following issues: what is the Parkinson Disease? why should we make any investigation at this direction? Why machine learning could be useful? Why should ‘non-specialist’ make timely identification of suitable candidates for DAT? etc.
2. You should describe the Parkinson’s Kinetigraph system in the paper using much more details (it is not sufficient to give references).
3. You should compare your results with results obtained by other authors.
Minor
1. The paper needs an extensive editing work. It contains of many additional and unnecessary spaces at the start of the sentences. Moreover, the first paragraph in the abstract section should be justified.
2. Line 121: “decision. they met both criterion 1 AND 2 (above)” – uppercase is needed. There are no easily identified (by means of numbers) criterions 1 and 2 given above. Please compare to lines 183-196.
3. Lines 129-130 – Is there any sense to create a subsection of one sentence long?
4. What is the ‘delta’ and p-value for age parameter in the table 1?
5. Figure 1a – if ‘ExplainedVariance’=eigenvalue?.
6. Figure 1a is a plot of the first 7 components. What does mean the component number 0?
7. Figures 1b and 1c – what does mean the pink triangles (diamonds)?
8. Lines 286-287. Which criterion has been used for the selecting of the number of principal components? How many of the variance do the first three components explain?
9. Line 624 – table 1 of supplementary results – please notice that delta for age parameter for construction population is empty.
Author Response
see attached word document

Reviewer 2 Report
This paper aimed to present a classifier that model specialist identification of suitable DAT candidates. The classifier’s performance was assessed in people with Parkinson's disease who underwent Deep Brain Stimulation. This technology provides evidence that information from objective measurement could improve timely referral for Device Assisted Therapies. The paper is interesting and well written; however, there are a few changes that could be made to make the results more accessible and clear to readers, in details:
1. The main points of the paper should be emphasized in the abstract section. The reader need more help to understand what is important, what is new, and how it relates to the state of art.
2. As for Figure 1, the authors claim that the first three components can explain most of the variance in the feature space. But they did not offer an explanation.
3. Wearable sensors are attractive for human motion tracking applications because these systems can measure and obtain real-time motion information outside of the laboratory for a longer duration. More literatures could be provided in the paper to get better coverage of the research field. Some examples are:
- Al-Amri, M., Nicholas, K., Button, K., Sparkes, V., Sheeran, L., & Davies, J. L. (2018). Inertial measurement units for clinical movement analysis: Reliability and concurrent validity. Sensors (Switzerland), 18(3), 1–29.
- Albert, M. V., Azeze, Y., Courtois, M., & Jayaraman, A. (2017). In-lab versus at-home activity recognition in ambulatory subjects with incomplete spinal cord injury. Journal of NeuroEngineering and Rehabilitation, 14(1), 1–6.
- Qiu, S., Wang, Z., Zhao, H., Liu, L., & Jiang, Y. (2018). Using Body-Worn Sensors for Preliminary Rehabilitation Assessment in Stroke Victims with Gait Impairment. IEEE Access, 6, 31249–31258.
4. In section V, authors give some figures. However, they do not make strong conclusions. What are the implications of the findings? More discussions could be added in the manuscript.
5. There are unnecessary uppercase throughout the manuscript.
Based on the above considerations, I recommend a minor revise.
Author Response
see attached word document

Reviewer 3 Report
1. Lack of novelty
I am concerned about the novelty of the work presented in the manuscript. The Authors have not provided the review of the state-of-the-art findings in the considered area, and therefore it is unclear whether similar research has already been done. If the Authors are pioneers in this area, this information should be clearly stated; however, if similar research is being conducted elsewhere, the Authors should point out the differences in their approach.
2. Methodology
How many specialist (mentioned in line 185) provided the reference data for the experimentation? In Section 5, in line 618, a “relatively small number of clinicians” is mentioned, but exact number is nowhere to be found. To maximise objectivity of the classification, the classifier should be trained on reference data from various clinics because a small group of co-workers may be similarly biased.
3. Figures
The captions of Figure 1, Figure 3 and Figure 4 contain new information, comments and conclusions: this information should be placed in the main text, while the captions should be made shorter.
4. Conclusions
This section is too short and lacks important information. Conclusions should contain summary of the experiments, their results and discussion, as well as information regarding the further work.
5. Language
The text of the manuscript is grammatically and stylistically unacceptable; my main objections are as follows:
5.1. The term Parkinsons Kinetigraph (appearing for the first time in the title) is spelled in three different ways throughout the manuscript: Parkinsons Kinetigraph (line 2), Parkinson’s Kinetigraph (line 53) and Parkinson’s KinetiGraph (line 632). Please unify the spelling.
5.2. The use of capital letters in the titles of the (sub)sections is rather random. Please use the journal recommendations.
5.3. British and American spelling is mixed.
5.4. There are a lot of terms that are needlessly written with capital letters, e.g. device-assisted therapies, construction population, test population, sensitivity, specificity, deep brain stimulation, classifier score, clinical “criteria negative”, clinical “criteria positive”, all of the PKG terms, ensemble learning, bagging, radial basis function and so on…
5.5. Throughout the manuscript, the Authors mention “machine learning model”, “machine learning system”, “classifier” and “classifier algorithm” interchangeably, while – in my opinion – they always refer to “classifier”. The terminology should be unified because it is confusing in its current form.
5.6. The majority of sentences are separated with a wide space. It should be corrected.
5.7. Often there is no space between the last word in a sentence and a square bracket indicating a reference (e.g. see line 36).
5.8. In Table 1 and Supplementary Table 1 please use the “en-dash” sign to indicate a range of numbers and a “minus” sign to indicate the negative numbers.
5.9. The sentences in lines 158–159 and 542–543 are unintelligible.
5.10. Other comments:
– Line 2: The correct spelling of “device assisted therapies” is “device-assisted therapies”.
– Lines 25–29: This sentence is not clear. Who/what agreed with what? Please use punctuation marks.
– Line 102: The correct spelling of “real world practice” is “real-world practice”.
– Line 121: The word “they” should start with capital letter.
– Line 126: The word “if” should start with capital letter.
– Line 138: The correct spelling of “wrist worn logger” is “wrist-worn logger”.
– Line 138: The correct spelling of “two minute period” is “two-minute period”.
– Line 147: The period at the end of the sentence is missing.
– Line 165: The period at the end of the sentence is missing.
– Line 173: The period at the end of the sentence is missing.
– Lines 206–209: The full names of the features are not necessary; the abbreviations from Section 2 will suffice.
– Lines 233–235: The full names of the features are not necessary; the abbreviations from Section 2 will suffice.
– Line 266: The correct spelling of “equal size subsamples” is “equal-size subsamples”.
– Line 424: The period at the end of the sentence is missing.
– Line 483: The correct spelling of “post treatment score” is “post-treatment score”.
– Line 536: Please use either a percentage or a fraction.
I strongly advise the Authors to carefully read the manuscript before the possible resubmission; a help from a native-speaking colleague is recommended.
6. Other comments
– Line 30: What does the keyword “wearing off” refer to?
– Line 35: What is DBS?
– Line 39: What is PD?
– Line 40: What are “fluctuations”? What is subject to those fluctuations?
– Line 43: What are “off” periods?
– Lines 66–67: A clinician is not a benchmark. Clinician provides the reference data.
– Line 87: In what sense the model was optimal? What criterion was used?
– Line 244: What do the Authors mean by “non-overlapping descriptive information”?
– Line 257: What is CPD?
– Lines 267–269: In this sentence a reference to an external source is needed.
– Lines 300–301: In this sentence a reference to an external source is needed.
– Line 447: What is MDS?
– Line 443: In what sense the symptoms were “optimally controlled”? What criterion was used?
– Line 463: What is MoCA?
– Line 502: What is DAT Readiness Score?
–      Line 589 and 592: What is Clinical Classifier Score?

Author Response
see attached word document

Reviewer 4 Report
This is a well written paper. Most of my comments pertain to the machine learning section.
Page 6: Fig 1 (a): what is the y axis in this figure? What are the units? Is it percentage variance?
Page 6: Fig 1 (b-c): Have the authors tried linear discriminant analysis, which is a supervised dimensionality reduction and classification method. Similarly using non-linear embedding (t-sne) might provide some insight.
Page 7: line 272: why did the authors choose these two classification methods? What was the rationale? Did they consider other methods such as random forests or deep networks?
Page 7: line 282: What kernel was used for the support vector classifier?
Page 9: figure 2 (d): the linear and non-linear classifiers both lead to similar performance. Can we infer that using a computationally low cost classifier will suffice?
Other comments:
Page 3: line 133: the accelerometer system produces data points every 2 minutes. Is that enough resolution for this kind of analysis? It is not clear if this is the sampling rate of the accelerometer (Raw data) or if this is the processed information from the algorithms. Please elaborate.
Page 3: line 139: briefly elaborate on the data analysis methods.
Author Response
see attached document

Round 2
Reviewer 3 Report
Since the Authors have answered all of my comments, and introduced some essential changes to the manuscript, I recommend accepting the manuscript for publication.